# BINDING LANGUAGE MODELS IN SYMBOLIC LANGUAGES

**Zhoujun Cheng**[*♠♡]   **Tianbao Xie**[*♠]   **Peng Shi**[△]   **Chengzu Li**[♠]   **Rahul Nadkarni**[♣]
**Yushi Hu**[♣]   **Caiming Xiong**[♥]   **Dragomir Radev**[★]   **Mari Ostendorf**[♣]
**Luke Zettlemoyer**[♣♦]   **Noah A. Smith**[♣♢]   **Tao Yu**[♠♣]
[♠]The University of Hong Kong    [♡]Shanghai Jiao Tong University    [♣]University of Washington
[♢]Allen Institute for AI [△]University of Waterloo [♥]Salesforce Research [★]Yale University [♦]Meta AI

## ABSTRACT

Though end-to-end neural approaches have recently been dominating NLP tasks in both performance and ease-of-use, they lack interpretability and robustness. We propose BINDER, a training-free neural-symbolic framework that maps the task input to a program, which (1) allows binding a unified API of language model (LM) functionalities to a programming language (*e.g.*, SQL, Python) to extend its grammar coverage and thus tackle more diverse questions, (2) adopts an LM as both the program parser and the underlying model called by the API during execution, and (3) requires only a few in-context exemplar annotations. Specifically, we employ GPT-3 Codex as the LM. In the parsing stage, with only a few in-context exemplars, Codex is able to identify the part of the task input that cannot be answerable by the original programming language, correctly generate API calls to prompt Codex to solve the unanswerable part, and identify where to place the API calls while being compatible with the original grammar. In the execution stage, Codex can perform versatile functionalities (*e.g.*, commonsense QA, information extraction) given proper prompts in the API calls. BINDER achieves state-of-the-art results on WIKITABLEQUESTIONS and TABFACT datasets, with explicit output programs that benefit human debugging. Note that previous best systems are all finetuned on tens of thousands of task-specific samples, while BINDER only uses dozens of annotations as in-context exemplars without any training. Our code is available at `https://github.com/hkunlp/binder`[1].

## 1 INTRODUCTION

Performance on natural language processing tasks is dominated by neural *end-to-end* systems that directly map inputs to outputs (Devlin et al., 2019; Liu et al., 2019; Lewis et al., 2020; Raffel et al., 2020, *i.a.*). These end-to-end approaches are flexible and easy-to-use while lacking interpretability and robustness. This stands in contrast to *symbolic* approaches that produce explicit intermediate representations such as logical forms, reasoning paths, or program code, which might then be executed to derive a final output (Zettlemoyer & Collins, 2005; Gulwani et al., 2017; Chen et al., 2019b, *i.a.*). The intermediate form produced by these the resulting execution makes them more robust to input changes. However, their semantic coverage is limited by the affordances of the grammar of the selected symbolic language (*e.g.*, not being able to handle *"North America?"* in Fig. 1), leading to failures on real-world diverse questions, and the intermediate form annotations require expert knowledge and researcher labour.

A few works (Andreas et al., 2016; Gupta et al., 2019; Khot et al., 2021; Zhu et al., 2022, *i.a.*) have been proposed to combine neural modules and symbolic languages (*neural-symbolic*) to leverage advantages of both approaches. However, they require the elaborate human design of the symbolic language and the calibration of corresponding neural modules to tackle problems in a specific domain with large training data. More specifically, most of these works propose a task-specific symbolic language and corresponding modules that cover only limited semantic phenomena in a specific task and domain. Therefore, new languages and neural modules have to be introduced when adapting them

---

* Equal contribution. Authors in alphabetical order. Work mainly done at the University of Hong Kong.
[1]More resources at `https://lm-code-binder.github.io/`.

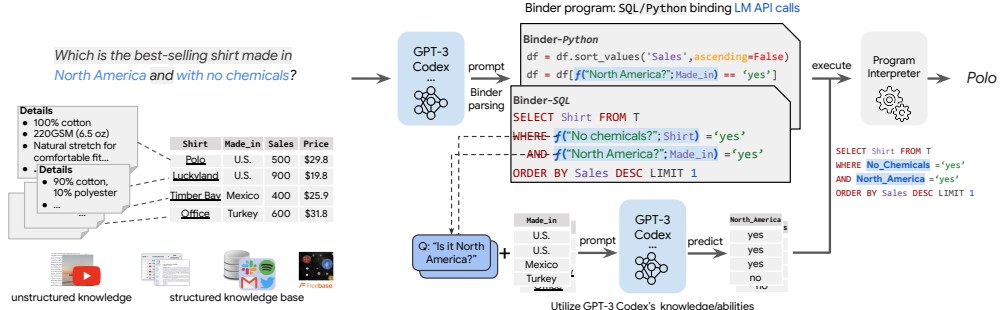

Figure 1: An overview of the BINDER pipeline of two stages: *parsing* and *execution*. (1) In the *parsing* stage, the language model (LM) maps the input to a BINDER program given the question and (optional) knowledge sources. The expressions with blue background in the program are API calls to acquire external results. (2) In the *execution* stage, an LM serves to realize the API calls given the prompt and the return values feed back into the original programming language. A deterministic program interpreter executes the program without API calls to derive the final answer.

to new tasks and domains. Their coverage is still restricted by the customized symbolic language and neural modules. Moreover, they call for various and large training data to ensure all modules are well trained. Therefore, we expect a neural-symbolic system that supports *flexible* neural module calls that will enable *higher coverage* for the symbolic language, while only requiring *few annotations*.

We propose BINDER, a training-free neural-symbolic framework that maps task inputs to an executable program in a programming language (e.g., SQL, Python) bound with a unified API to call language models (LMs; Brown et al., 2020; Chen et al., 2021) to perform versatile functionalities, *i.e.* a BINDER program(*e.g.*, *Binder-SQL, Binder-Python* in Fig. 1), with only a few input-BINDER program annotations as in-context exemplars. More specifically, BINDER first prompts Codex, a code-pretrained of GPT-3, to parse a question input into a BINDER program, in which Codex has to decide (1) which parts in the input can be converted to the target programming language (question parts highlighted in grey in Fig. 1), (2) the corresponding task API calls (*e.g.*, $f(\text{``North America?''; } Made\_in)$) to prompt Codex to resolve the other parts, and (3) where to insert the API calls in the BINDER program. Next, BINDER prompts Codex again to generate answers to the task API calls (given the generated task prompts), integrates the generated results back to the programming language. Specifically as in Fig. 1, the prompt (*e.g.*, "North America?") and data (*e.g.*, column $Made\_in$) in API calls are fed into Codex, and the output is a new column answering the prompt based on the input data (*i.e.*, yes/no of whether a country in $Made\_in$ column is from North America). Finally, the program with standard programming languages is executed the derive the final answer.

In summary, BINDER enables flexible functionality integration to the programming language to improve its coverage and requires only a few annotations. BINDER program replaces custom neural modules and task-specific languages with a unified prompting API call to Codex and general programming languages, respectively, to handle much more diverse task inputs in open domains without complex language and neural module design. BINDER is built on the advances of in-context learning with language models and does not require any training and large-scale annotations.

We demonstrate the effectiveness of the BINDER framework on WIKITABLEQUESTIONS (WIKITQ; Pasupat & Liang, 2015) TABFACT (Chen et al., 2019a), two structured knowledge grounding datasets that require complex reasoning on the tables. Using Codex (Chen et al., 2021) as the LM, BINDER achieves state-of-the-art results on WIKITQ and TABFACT. Note that the previous state-of-the-art methods all require fine-tuning on more than 10K annotated training examples or even massive amounts of task-related pretraining data, while our method requires only a dozen or so annotations without training. In further analysis, we find that BINDER provides the greatest performance gain on the questions that the original language grammar (SQL and Python) cannot support, indicating that BINDER effectively improves programming language coverage. We also demonstrate BINDER can be applied on multi-modal knowledge sources (text, table, images, and combined) with MUL-TIMODALQA dataset (Talmor et al., 2021). Moreover, we show that BINDER, compared with end-to-end approaches, is more interpretable when debugging the model, more scalable to very large inputs, and more robust to noisy inputs.

## 2 APPROACH

**Task Definition**   Given an NLP task that accepts a natural language question/statement $Q$ and optional context(s) $D$ (*e.g.*, passages, tables, images, or a combination of the above) as inputs, the goal is to output an answer $A$ based on the inputs to respond to $Q$ correctly. For example, in passage question answering, $Q$ is a question about the passage(s) $D$; in table fact verification, $Q$ is a statement about the table(s) $D$.

### 2.1 BINDER FRAMEWORK

**Overview**   The BINDER framework to solve NLP tasks is defined as follows: given a natural language input $Q$ and optional context(s) $D$ as the input, an *executable* BINDER program $Z$ is generated. Finally, the output answer $A$ is derived by executing $Z$ with a BINDER interpreter.

**BINDER Parsing**   In the parsing stage, the input natural language $Q$ is parsed into a BINDER program $Z$. A BINDER program is an expression in a symbolic language that optionally includes API calls where the core symbolic language fails to provide a desired functionality. We define the API call in the program as function $f(\hat{Q} ; \hat{D})$ that accepts a question $\hat{Q}$ to be answered, and the context $\hat{D}$ to be queried on. Here $\hat{Q}$ is the unanswerable part of $Q$ with the programming language only and $\hat{D}$ is the relevant contexts in $D$ to answer $\hat{Q}$. For example, *"North America?"* in Fig 1 is a $\hat{Q}$, and its corresponding contexts $\hat{D}$ to answer $\hat{Q}$ is the column *Made_in*. Note $\hat{Q} = Q$ or $\hat{D} = D$ is also valid (if $\hat{Q} = Q$ and $\hat{D} = D$, the program is equivalent to solving the problem with an LM in end-to-end manner). The output of an API call $f(\hat{Q} ; \hat{D})$ is the answer to $\hat{Q}$, and it is represented as *a variable* compatible with the symbolic language grammar so that the program can be executed.

**BINDER Execution**   In the execution stage, the program $Z$ is executed by a BINDER interpreter to derive the answer $A$. The BINDER interpreter consists of a standard symbolic language interpreter and the model(s) realizing the API calls. The execution phase includes lexical analysis, syntax analysis, and program evaluation. In lexical and syntax analysis, $f(\hat{Q} ; \hat{D})$ is added as a new identifier in the grammar, and the program is parsed as an abstract syntax tree (AST) based on this extended grammar. In program evaluation, the API calls are evaluated by calling the underlying neural models. The API call output is saved as *a variable* compatible with the standard symbolic language grammar, and thus the program can be finally executed by an off-the-shelf symbolic language interpreter to derive the output. Specifically, BINDER extends the symbolic language production rules to facilitate parsing the BINDER program's AST. The AST is evaluated in bottom-up order so that nested API calls are also supported, which allows different degrees of decomposition and free combination of language models for the complex question. We provide more details about the grammar extension and a BINDER program AST in Appendix A.4.

### 2.2 IN-CONTEXT LEARNING FOR BINDER

Much recent work uses large language models for in-context learning (Brown et al., 2020; Chen et al., 2021; Chowdhery et al., 2022b). Compared with fine-tuning, in-context learning (1) only takes a few annotations/demonstrations as a prompt, and (2) performs inference without training the model parameters, which are both longstanding issues of conventional semantic parsing. We base our experiments for BINDER on Codex, a code-pretrained version of GPT-3, which has been shown to perform proficiently on code generation tasks (Rajkumar et al., 2022; Chen et al., 2022) and even some weakly-related tasks like dialogue state tracking (Hu et al., 2022). We use Codex as both the semantic parser and the model to perform API call functionalities.

In the parsing stage, it is challenging to generate BINDER programs because their grammar is different from the original programming language grammar due to the inserted API calls. Thus, we take advantage of the few-shot generalization ability of Codex and find that it can learn the modified grammar effectively with only a small number of in-context examples. In the execution stage, Codex as the underlying LM serves to give the output to the API calls by concatenating the API call input, $\hat{Q}$ and $\hat{D}$ as the language model prompt (the prompt style is described in Section 3.1). The Codex output result(s) are stored as *variables* in the standard programming language so that a programming

language interpreter can execute on the combination of these variables and the rest part of the program.

Specifically, we apply in-context learning for BINDER in the following manner: the inputs are $k$ in-context exemplars of $\{(Q_i, D_i, Z_i)\}_{i=1}^{k}$ and the inference example $(Q, D)$. The $k$ examples should balance the trade-off between the diversity of question types and the model's maximum input capacity, which can either be manually selected (fixed) or automatically retrieved (dynamic) according to the inference example. The model outputs are $n$ candidate BINDER programs $\mathcal{Z} = \{Z_1, ..., Z_n\}$ that aim to solve the inference question $Q$. Next, the programs $\mathcal{Z}$ are executed by the BINDER interpreter producing $n$ answers $\mathcal{A} = \{A_1, ..., A_n\}$. Finally, the output answer $A$ is derived via a majority voting strategy over the set of produced answers $\mathcal{A}$ (the voting method is similar to the one used by MBR-EXEC (Shi et al., 2022), which we elaborate on in Appendix A.3). The values of $k$ and $n$ are hyperparameters in the process above.

## 2.3 BINDER IMPLEMENTATION

In this section, we describe our implementation of BINDER with SQL and Python over structured knowledge as a demonstration. As BINDER is designed to be extensible to various programming languages and API call functionalities, we introduce a pipeline for users to quickly adapt BINDER to a new domain in Appendix D.

We implement two APIs — $f_{col}(\hat{Q}\,;\hat{D})$ and $f_{val}(\hat{Q}\,;\hat{D})$, where $\hat{D} = \{c_1, ..., c_{|\hat{D}|}\}$ is a (sub-)table of a set of table columns, and $c = \{v_i, ..., v_{|c|}\}$ is a column filled with cell values. Based on $\hat{Q}$, $f_{col}$ maps $\hat{D}$ into *a column*, and $f_{val}$ maps $\hat{D}$ into *a value*. Since both return types, i.e., column and value, are compatible with the grammars of SQL and Python (with the Pandas package), $f_{col}$ and $f_{val}$ APIs are inserted to replace the columns and values to form a valid BINDER program.

Take the question "*which is the best-selling shirt made in North America and with no chemicals?*" in Fig 1 as an example. The country names in the *Made_in* column do not provide enough information on their own to indicate what their continents are, and thus pure SQL cannot solve it (with this table's data). However, it is easy for a (large) language model to answer whether a country is from North America, which is represented by the $f_{col}(\text{"North America?";Made\_in})$ expression in the position of a column name in standard SQL. Similarly, the expression $f_{col}(\text{"No chemicals?";Shirt})$ calls a (large) language model to identify whether the shirts consist of no chemicals (*i.e.*, pure cotton in this case) based on the textual details in the *Shirt* column.

When a (sub-)question is too complex or infeasible to be solved by creating an intermediate new column with $f_{col}$, we turn to $f_{val}$ to directly derive the answer. For example, given the question "*which shirt is the most suitable for a formal event?*" on the table in Fig 1, it is hard to map *Shirt* to a new column of "formality value" followed by a SQL "ORDER BY" clause. Thus, the expression will be $f_{val}(\text{"The most formal?";Shirt})$, that outputs a value as the answer. $f_{val}$ looks more like end-to-end QA, with two important differences: (1) it can be integrated into more compositional SQL queries using its result as a value, (2) it inputs the sub-table instead of the whole table which can mitigate the challenge of input capacity.

# 3 EXPERIMENTS

## 3.1 EXPERIMENT SETUP

**Datasets** We evaluate our method on three knowledge grounding datasets which were all previously dominated by *end-to-end* methods: WIKITQ (Pasupat & Liang, 2015) and TABFACT (Chen et al., 2019a). WIKITQ requires complex table reasoning skills to answer the questions. Furthermore, according to SQUALL (Shi et al., 2020b), about $20\%$ of WIKITQ questions are not answerable by pure SQL, either because of the need for extra knowledge or the limited coverage of the SQL grammar, both of which are issues that BINDER is designed to address. TABFACT is a binary fact verification benchmark over small tables, on which end-to-end methods have a large advantage but offer no interpretability.

| Method | Dev. | Test |
|---|---|---|
| *Finetuned* | | |
| T5-3B (Xie et al., 2022) | 51.9 | 50.6 |
| Tapex (Liu et al., 2021) | 60.4 | 59.1 |
| TaCube (Zhou et al., 2022) | 61.1 | 61.3 |
| OmniTab (Jiang et al., 2022) | - | 63.3 |
| *Without Finetuning* | | |
| Codex end-to-end QA | 50.5 | 48.7 |
| Codex SQL$^{\dagger}$ | 60.2 | 61.1 |
| **Codex BINDER $^{\dagger}$ (Ours)** | **65.0** | **64.6** |

Table 1: WIKITQ execution accuracy on development and test sets. $\dagger$ denotes a symbolic method that outputs intermediate languages.

| Method | Test |
|---|---|
| *Finetuned* | |
| SASP$^{\dagger}$ (Ou & Liu, 2022) | 77.0 |
| BART-Large (Lewis et al., 2020) | 82.5 |
| T5-3B (Xie et al., 2022) | 85.4 |
| Tapex (Liu et al., 2021) | **85.9** |
| *Without Finetuning* | |
| Codex end-to-end QA | 72.6 |
| Codex SQL$^{\dagger}$ | 80.7 |
| **Codex BINDER $^{\dagger}$ (Ours)** | **85.1** |
| with few-shot retriever | **86.0** |

Table 2: TABFACT accuracy on the official small test set. $\dagger$ denotes a symbolic method that outputs intermediate languages.

**Evaluation**   The evaluation metrics are execution accuracy (EA) for WIKITQ and TABFACT following common practice for these datasets. Program executions are likely to be semantically correct but fail to match the gold answer exactly – for example, SQL outputs 1/0 for yes/no questions. Though this is considered correct according to human evaluation, it is regarded as incorrect by the exact match evaluator. Thus, we add a pre-matching check for these semantically correct cases in WIKITQ to the official evaluator. For a fair comparison, we compute the outputs of all baseline methods and re-evaluate them using the same evaluator. We provide more details on the evaluator in Appendix A.5, and list the results with the official evaluator for all baselines and our method in C.2.

**Baselines**   We compare our method to a series of strong published methods on these datasets, including Tapex (Liu et al., 2021), OmniTab (Jiang et al., 2022), TaCube (Zhou et al., 2022), T5-3B (Xie et al., 2022; Raffel et al., 2020), and SASP (Ou & Liu, 2022). These baselines are fine-tuned on the full-size training set, and many of them (Liu et al., 2021; Zhou et al., 2022) are even further pretrained on a domain-relevant corpus of extra data which is specific to the target domain and task, while BINDER is training-free and only requires a few in-context exemplar annotations.

To further demonstrate the effectiveness of BINDER, we also evaluate Codex with additional inference modes including: (1) end-to-end QA, i.e., directly outputting the answer based on the input question and table; and (2) semantic parsing with the standard SQL language. Due to page limits, we refer readers interested in these baselines to their respective papers for details (Liu et al., 2021; Jiang et al., 2022; Zhou et al., 2022; Xie et al., 2022; Raffel et al., 2020; Ou & Liu, 2022), and we present a complete list and evaluation results of more previous systems on these datasets in Appendix C.1.

**Implementation Details**   We use the OpenAI Codex (code-davinci-002) API[2] model in our experiments as both the parser to generate programs and as the underlying model for API calls during the execution of each program. We annotate 14 in-context exemplars with BINDER programs for each dataset, which are selected considering the diversity of question types in demonstrations and the maximum token limit for Codex ($8, 000$ tokens). The prompt format mainly follows (Rajkumar et al., 2022), which inputs the table schema and the first three table rows. The detailed prompt templates we use for each dataset are listed in Appendix A.1. On TABFACT, we further annotate a pool of $200$ examples with BINDER programs from the training set using vote-$k$ selective annotation (Su et al., 2022), and use them by retrieving relevant few-shot examples for each inference example via maximum inner-product similarity of the SentenceBert (Clark et al., 2019) embeddings of the questions. The Codex hyperparameters for parsing and execution are provided in Appendix A.2. Empirically, it takes about 6 hours on average to evaluate $1, 000$ examples (i.e., generate and execute 20 programs per example) given that Codex allows 200 queries per minute. We evaluate on the official small test set ($2, 000$ samples) of TABFACT considering the time cost.

## 3.2   MAIN RESULTS

**WIKITQ**   All results on the WIKITQ dataset are shown in Table 1. Our Codex BINDER outperforms listed strong baseline systems by a large margin, surpassing the previous state-of-the-art by absolute

---

[2]https://beta.openai.com/

| Method | Program-unsolvable | Program-solvable | Overall |
|---|---|---|---|
| T5-3B (Xie et al., 2022) | 37.6 | 56.0 | 51.9 |
| Tapex (Liu et al., 2021) | 33.6 | 68.0 | 60.4 |
| TaCube (Zhou et al., 2022) | 34.9 | 68.5 | 61.1 |
| Codex end-to-end QA | 40.3 | 53.4 | 50.5 |
| *w/o table inputs* | 14.2 | 11.9 | 12.4 |
| Codex SQL | 31.2 | 68.4 | 60.2 |
| **Codex BINDER (Ours)** | **41.3** | **71.8** | **65.0** |

Table 3: Decomposition of execution accuracy on WIKITQ development set. The questions annotated with SQL by SQUALL (Shi et al., 2020b) dataset are denoted *program-solvable*, and the rest are *program-unsolvable*. OmniTab (Jiang et al., 2022) didn't provide its result on the development set.

1.3%. Note that all baseline methods require access to the full training dataset and further fine-tuning, while our method can achieve state-of-the-art performance with only 14 example annotations. Codex BINDER improves over semantic parsing with standard SQL by 3.5% on the test set, indicating BINDER indeed mitigates the coverage limitations of the original language. Furthermore, both Codex BINDER and Codex SQL deliver dramatic advantages (15.9% and 12.4%) over end-to-end QA, showing that semantic parsing with Codex is a better default choice when it comes to structured knowledge grounding and code-related tasks. A fine-grained analysis on why BINDER outperforms end-to-end QA and pure SQL is provided in Section 4.1.

**TABFACT** Table 2 presents the results on TABFACT's official small test set. Codex BINDER substantially surpasses the previous best symbolic method SASP by 8.1%. Note symbolic methods usually fall behind end-to-end manners in fact verification (binary classification) since the answer space is small. When retrieving a few relevant questions from the annotated pool of 200 examples, our method achieves new state-of-the-art results, outperforming the previous best fine-tuned method. However, the improvement provided by retrieving a few similar examples is comparatively small (0.9%). Effective sample selection methods for in-context learning remain an open challenge. Within the Codex few-shot setting (no few-shot retriever), BINDER shows a large advantage over standard SQL (4.4%) and end-to-end QA (12.5%), indicating the necessity of BINDER framework.

Besides the performance gain, BINDER has the additional advantages of (1) interpretability that benefits human debugging and (2) robustness that makes it stable to large or noisy inputs. We elaborate on these in Sections 4.2 and 4.3.

# 4 ANALYSIS

## 4.1 ABLATION STUDY

Binding neural module API calls into a programming language can help solve queries that are unsolvable in that language alone. Therefore, we are particularly interested in the performance of BINDER on the unsolvable questions. According to SQUALL (Shi et al., 2020b), about 20% of WIKITQ questions cannot be annotated in SQL; we call these *program-unsolvable*, and refer to the annotated ones as *program-solvable*. As presented in Table 3, Codex BINDER significantly outperforms Codex SQL by 10.1% on *program-unsolvable* questions, aligned with the motivation of our design. Our method even performs better than all end-to-end QA methods on the *program-unsolvable* part, which are supposed to be more robust to this subset of the examples. We note that while SQL performs well on about 31.2% of unsolvable questions, many are spurious programs that derive the correct answer by accident. BINDER largely mitigates this phenomenon. We randomly pick 100 correct predictions in *program-unsolvable* set shared by SQL and BINDER and find that BINDER has a much lower spurious rate than SQL (12% vs. 33%). One thing to note is that BINDER also achieves a higher score than SQL on *program-solvable*. We find that this is because some tables are manually cleaned (*e.g.*, extracting numerical parts from the text) to run the annotated SQLs by SQUALL, which means there also exist unsolvable questions even in the *program-solvable* subset. We elaborate on our manual statistics about the proportion of WIKITQ that can be solved with SQL in Appendix C.4. We also present to what extent Codex itself can answer the question only using its internal knowledge. Codex can only answer a few WIKITQ questions (12.4%) correctly, indicating it is not pretrained to overfit to the downstream dataset questions.

| Error Type | Description | Proportion(%) |
|---|---|---|
| Syntax error | Incorrect program syntax (invalid grammar) | **5%** |
| Semantic error | Incorrect program semantics | **64%** |
| Token | Incorrect or missing column/value/operator | 10% |
| Structure | Incorrect program structure (valid grammar) | 22% |
| BINDER usage | Missing BINDER API usage | 32% |
| Incorrect execution | Incorrect execution answer with the correct program | **15%** |
| False negative | Incorrect annotations or misjudge in evaluator | **16%** |

Table 4: Error types of 100 samples from WIKITQ development set of BINDER (SQL).

| System | Program | Answer |
|---|---|---|
| End-to-end QA | – | *1967* |
| SQL | `SELECT year FROM t WHERE win_team='kansas state' AND win_team – los_team > 10` | *<empty>* |
| BINDER | `SELECT year FROM t WHERE win_team='kansas state' AND f("Points?";win_team) – f("Points?";los_team) > 10` | *<empty>* |

Table 5: An example from WIKITQ development set. The query is "*When was the first game that kansas state won by double digits?*" and the gold answer is 1926. The incorrect segment(s) of each output are marked in red. See Appendix E for full context of this example.

## 4.2 INTERPRETABILITY

An important advantage BINDER provides is the improvement in interpretability over the end-to-end approaches, where the explicit program can assist human debugging and error analysis. We sample 100 error cases from WIKITQ dev set (approximately 10% of all incorrect examples) of the Codex BINDER (on SQL) for error analysis. We classify errors into syntax error, semantic error, incorrect execution, and false negative, as listed in Table 4. More details about the classification and example demonstrations are listed in Appendix E. For WIKITQ, the errors mainly lie in the BINDER usage (32%) and structure errors (22%) of semantic errors, and incorrect execution (15%). This indicates that 32% of incorrect examples can be further corrected if BINDER is used and 15% can be corrected through better execution, *e.g.*, by leveraging more powerful LMs, using better in-context learning methods, and annotating more exemplars.

We also use an example to show the advantages of using BINDER for debugging and interpreting results. As shown in the example in Table 5, the result from the end-to-end system is *1967*, which is incorrect but provides no clue to the reason behind this prediction. The result from BINDER is incorrect, since the program uses the incorrect value and operator regarding the *win_team* column from the table (should be "LIKE "%kansas state%""), and it can be potentially fixed in the future by adding similar in-context exemplars or fuzzy match postprocessing. For standard SQL, though we find the direct subtraction of *win_team − los_team* is incorrect (contain non-numerical text), it cannot be fixed for its limited grammar coverage. Thus, in this example, BINDER enables finding the source of the error (an advantage over the end-to-end approach) while also being expressive enough for users to find a way to fix it (an advantage over a pure symbolic system), making BINDER fit for debugging and enabling interpretability of a system that uses it. We leave it to future work to develop more systematic evaluation methods that consider ease of debugging by estimating human effort to correct system errors.

## 4.3 ROBUSTNESS

### 4.3.1 SCALABILITY

A great advantage of BINDER over end-to-end QA is the scalability, *i.e.*, parsing and executing the symbolic language is always practical even when the knowledge source is too large (*e.g.*, company databases, domain knowledge graphs) to fit into the model input capacity or memory, while end-to-end QA fails or degrades since it requires the whole table as input to predict the answer. We expand tables with 100, 200, and 500 rows from a WIKITQ development subset (random 150 samples) by prompting Codex to populate the original table with content-consistent rows, and then have two annotators annotate question-answer pairs manually. When the table is too large for Codex's maximum token limit (8,000 tokens), we truncate table rows until it fits. As shown in Figure 2, Codex

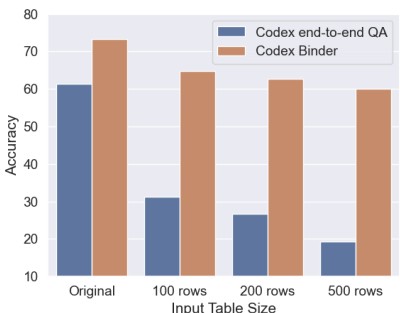

Figure 2: Execution accuracy on WIKITQ with very large tables (original, 100, 200, 500 rows).

Figure 3: Execution accuracy on WIKITQ with noisy content in tables.

| Method | Program-unsolvable |
|---|---|
| Codex Python | 30.7 |
| **Codex BINDER (Ours)** | **34.4** |

Table 6: WIKITQ accuracy on program-unsolvable subset using Python as the programming language.

| Method | F1 | EM |
|---|---|---|
| *Finetuned* | | |
| Implicit-Decomp (Talmor et al., 2021) | 55.5 | 48.8 |
| PReasM-Large (Yoran et al., 2022) | **65.5** | **59.0** |
| *Without Finetuning* | | |
| Codex end-to-end QA | 55.4 | 48.0 |
| **Codex BINDER (Ours)** | 57.1 | 51.0 |
| *with oracle retriever* | **64.5** | **58.1** |

Table 7: MMQA F1/EM on development set.

end-to-end QA performance drops dramatically as table size increases, while BINDER consistently outperforms it with only slight performance decreases. Note that the input of BINDER is only *three table rows* for all table sizes.

### 4.3.2 NOISY CONTENT

End-to-end methods are more brittle to noisy inputs, especially when there exists *distractors* that are similar to the question-relevant (gold) contents. We build a noisy WIKITQ development subset (random 150 samples) with distractors in three steps: (1) the gold table cells and columns are found using a heuristic fuzzy match, (2) replace 15% cells in gold columns with either *text* that has the smallest edit distance to the gold cell string in the WordNet corpus or *number* that equals to the gold cell value $\pm 1$, (3) have one annotator check that question-answer pairs are valid after content disturbance. As shown in Figure 3, BINDER is stable confronting distractors (1.3% ↓), while end-to-end QA is more likely to be confused by similar text and numbers (6.7% ↓).

### 4.4 BINDER EXTENSION

**BINDER with Python** We design BINDER to be easily extensible to various programming languages. Thus, we explore using Python (with the Pandas package) as the BINDER language on WIKITQ. Similar to SQL, BINDER with Python is implemented by incorporating the $f(\cdot\,;\cdot)$ neural API into it. Since the neural API also calls language models with Python, we just use the original Python interpreter to execute our BINDER with Python. We evaluated it on the program-unsolvable subset of WIKITQ to test whether our method improves Python's capability. Though this subset is split based on SQL, Python shares some similar weaknesses with SQL, such as in cases with external knowledge requirements and unsupported functionality. As shown in Table 6, BINDER with Python effectively improves the Python coverage on the difficult subset. The gap between Python and SQL performance on WIKITQ may lie in the properties of each programming language, as suggested by Guo et al. (2020). We conjecture that Codex is more familiar with semantic parsing to SQL for tabular data, while Python is not as commonly used in this context.

**MultiModal Application** We further explore applying BINDER on the multi-modal dataset MULTIMODALQA (MMQA) across text, tables, and images. To input images into the LM program generation, images are converted into textual image captions with a vision-text pretrained model OFA (Wang et al., 2022) in advance. For table-related questions, the BINDER program is based

on SQL. For non-table questions, a program with $f_{val}$ is generated with the targeted passage or image title(s). We follow Yoran et al. (2022) to retrieve the question-relevant passages and images in preprocessing. As far as we know, we are the first to demonstrate that multi-modal QA across text, tables, and images can be handled with interpretable and executable programs. As listed in Table 7, under the Codex few-shot setting, BINDER achieves better performance than end-to-end QA and the fine-tuned baseline Implicit-Decomp, showing the feasibility of BINDER on multi-modal knowledge sources. With the oracle retriever that assumes gold passages and images are given for each question, BINDER can achieve comparable performance with the state-of-the-art, showing the potential of BINDER approach in the future.

## 5 RELATED WORK

**Semantic Parsing**    Semantic parsing (Zelle & Mooney, 1996; Zettlemoyer & Collins, 2005) has been a mainstay of symbolic methods that produce an executable program given natural language input, generate intermediate structures that assist problem-solving, and improve interpretability over the neural methods which generate solution directly that came later. Structured knowledge grounding tasks mainly adopt semantic parsing since symbolic languages like SQL, SPARQL can be executed on them (Berant et al., 2013; Liang et al., 2017; Yin & Neubig, 2017; Zhong et al., 2018; Yu et al., 2018; Shaw et al., 2021; Scholak et al., 2021). Beyond structured knowledge, Chen et al. (2019b) and Thorne et al. (2021) design domain-specific languages executable on text. Recently, Chen et al. (2021) propose a generative pre-trained language model for code generation that require no additional human annotation. Many works propose methods based on this model and achieve great success (Rajkumar et al., 2022; Shi et al., 2022). However, the semantic parsing method is restricted by its grammar coverage, unable to solve problems requiring external knowledge or functions.

**Neural-Symbolic Methods**    Some works integrate neural modules with symbolic languages for the advantage of both approaches, *i.e.*, good performance and interpretability. Andreas et al. (2016); Hu et al. (2017); Das et al. (2018) and Gupta et al. (2019) generate programs that are further softly executed by the corresponding neural modules. Khot et al. (2021) propose text module networks to solve complex tasks by decomposing them into simpler ones solvable by existing QA models and a symbolic calculator. BREAK (Wolfson et al., 2020) proposes a meaningful representation, QDMR, that decomposes the question into multiple steps. However, they require the elaborate design of functions to be used in corresponding task and the calibration of corresponding neural modules which require complicated training steps and large training data (normally tens of thousands) to tackle problems in a specific domain.

Compared with these methods, BINDER is expressive and flexible to handle real-world diverse questions since it is able to make proper API calls to enhance its functionalities. Moreover, BINDER is training-free and requires only dozens of annotations based on certain symbolic language to perform on a domain specific task while maintaining: excellent performance, ability in scaling on input, interpretabilityand robustness over noisy content.

## 6 CONCLUSION

We propose BINDER, a training-free neural-symbolic framework that maps the task input to a program that allows binding a unified LM API for additional functionalities. BINDER aims to combine the strengths of end-to-end approaches (high coverage) and symbolic approaches (high interpretability). Using Codex as the LM, BINDER achieves state-of-the-art performance on WIKITQ and TABFACT with only dozens of in-context demonstrations and no additional training. In contrast, the best existing systems are finetuned on thousands of task-specific training samples and may require further domain-specific pertaining. We also perform a series of analyses of BINDER, decomposing performance gains, examining robustness to large or noisy inputs, applying it to multi-modal knowledge sources, and extending it to the Python language. We regard BINDER as a new, language model-focused attempt to integrate two widely-adopted paradigms in NLP: end-to-end and symbolic approaches. With the recent powerful large language models, it has become feasible that BINDER programs are generated and executed correctly in a training-free manner. In the future, we believe BINDER can be extended to many more scenarios with the appropriate programming language and functionalities, and we hope it can inspire more creative ideas on balancing model capability and interpretability.

## 7 REPRODUCIBILITY

BINDER experiments are mainly based on OpenAI Codex (code-davinci-002) API[3]. We provide (1) the input prompt templates we use in Appendix A.1, (2) Codex hyper-parameters for each dataset we adopt in Appendix A.2, (3) more implementation details in Appendix A. Besides, we also upload our source code in the materials making our results easy to be reproduced.

---

[3]`https://beta.openai.com/`

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

# Appendices

## A    MORE IMPLEMENTATION DETAILS

We present more details in our implementation of Codex BINDER for interested readers.

### A.1    CODEX INPUT PROMPTS

We list prompts we use in main experiments for BINDER for each dataset. As shown, the prompt starts with an instruction describing the task, followed by a few examples for in-context learning. We follow Rajkumar et al. (2022) table prompt format: (1) "CREATE TABLE" schema; (2) the first three rows of the table; (3) the question $Q$ and parsed logic form BINDER program. Besides, we also add *row_id* and lower case all table contents, which we find will improve the Codex parsing performance. For the sample to be inferenced, we empty the BINDER program to let Codex generate it, and input the full table to provide more information. If the full table is too large, we will shrink number of in-context shots until the table fits within the Codex max token limits.

```
## WikiTQ
Generate SQL given the question and table to answer
the question correctly.
...

CREATE TABLE Electoral district of Lachlan(
    row_id int,
    member text,
    party text,
    term text)
/*
3 example rows:
SELECT * FROM w LIMIT 3;
row_id member party term
0 john ryan none 1859-1864
1 james martin none 1864-1869
2 james watson none 1869-1880
*/
Q: of the members of the third incarnation of the lachlan,
    who served the longest?
Binder: SELECT member FROM w ORDER BY
    f("How long does it last?"; term) DESC LIMIT 1

## TabFact
Generate SQL given the statement and table to verify
the statement correctly.
...
CREATE TABLE british records in athletics(
    row_id int,
    event text,
    data text,
    athlete text,
    date text,
    place text)
/*
3 example rows:
SELECT * FROM w LIMIT 3;
row_id event data athlete date place
0 5 km t19:29 andi drake 1990-05-27 00:00:00 søfteland , norway
```

```
1 5 miles 32:38 + ian mccombie 1985-03-23 00:00:00
york , united kingdom
2 10 km 40:17 chris maddocks 1989-04-30 00:00:00
burrator , united kingdom
*/
Q: there be 8 different event that take place within
    the united kingdom
Binder: SELECT (SELECT COUNT(place) FROM w WHERE
    f("Is it in united kingdom?"; place) = 'yes') = 8

## MMQA
Generate SQL given the question, table, passages, image captions
to answer the question correctly.
...
CREATE TABLE 2018 Warrington Wolves season(
    row_id int,
    player text,
    signed from text,
    contract length text,
    announced text)
/*
3 example rows:
SELECT * FROM w LIMIT 3;
row_id player signed from contract length announced
0 sitaleki akauola penrith panthers 2 2017-08-01
1 bryson goodwin south sydney rabbitohs 2 2017-10-01
2 tyrone roberts gold coast titans 3 2017-10-01
*/
CREATE TABLE Images(
    row_id int,
    gold coast titans text)
/*
All rows of the table:
SELECT * FROM w;
row_id gold coast titans
0 a logo for the golden knights is painted on the beach.
*/
Q: What player was transferred from the team that has crossed
    swords on its logo to the Warrington Wolves in the 2018 season?
Binder: SELECT player FROM w WHERE
    f("Has crossed swords on its logo?"; `signed from`) = 'yes'
```

In end-to-end QA setting, the prompt is almost the same, except the full table contents are used in all the in-context examples to since in most cases QA must see the complete table to answer the question correctly. We empirically find that full table input to Codex is a necessity for end-to-end QA, but only a small bonus for semantic parsing.

In the ablation study experiments, the prompt is also almost the same, except that under the very large table *scalability* setting, we truncate number of table rows instead of in-context shots to fit in Codex max token limits, in order to simulate a realistic situation where only parts of the table can be input into the model.

## A.2 CODEX HYPER-PARAMETERS

In both two phases of our Codex BINDER pipeline, parsing and execution, the Codex api is called. We set the Codex in-context learning hyper-parameters as shown in Table 8.

| Hyper-parameter | Parsing | | | Execution | | |
|---|---|---|---|---|---|---|
| | WIKITQ | TABFACT | MMQA | WIKITQ | TABFACT | MMQA |
| temperature | 0.4 | 0.6 | 0.4 | 0.0 | 0.0 | 0.0 |
| top_p | 1.0 | 1.0 | 1.0 | 1.0 | 1.0 | 1.0 |
| max_output_tokens | 512 | 512 | 512 | 1024 | 1024 | 1024 |
| sampling_n | 20 | 50 | 20 | 1 | 1 | 1 |
| stop_tokens | \n\n | \n\n | \n\n | \n\n | \n\n | \n\n |
| num_shots | 14 | 14 | 18 | 8 | 8 | 8 |

Table 8: Codex hyper-parameters we set in main experiments. Codex is used in two phases: *Parsing* and *Execution*.

**Parsing**   In parsing phase, *i.e.* parsing questions into programming languages, we annotate a dozens or so examples from training set or modified examples from development set as in-context demonstrations for each dataset. $temperature$ is to control randomness of generations. We find $0.4$ is a suitable trade-off between fidelity and diversity of generated programs in most cases. $sampling\_n$ is number of generations (programs) per sample. We heuristically set $sampling\_n$ as 20 in WIKITQ and MMQA to balance the time cost in execution and the potential performance gain over majority voting. For TABFACT, we increase $temperature$ to 0.6 and 50, as we empirically find that for binary classification tasks (or tasks with a small output space), more generations with more diversity can better determine the final answer.

**Execution**   As mentioned in Section 2, the executor will call model(s) with various functionalities. In experiments, we use Codex as the underlining model for all the neural modules since Codex itself is a very powerful large language model which can realize most functions on texts and codes. For images, they are preprocessed into text via image captioning by OFA (Wang et al., 2022). Note that on MMQA, replacing Codex (with image captions) with a specific VQA model is likely to improve the performance on images because image captioning is question-agnostic while VQA model can predict the answer based on the question.

We let Codex output only one answer with $temperature$ 0.0 per input instead of multiple answers followed by a majority vote considering the time cost. An interesting thing is that even when $temperature$ is 0.0, the Codex outputs may be different in two inferences. OpenAI team said the randomness inherent to GPU computations, and generally the outputs will be the consistent in most cases, which is aligned with our observations.

We also find it better to give Codex some in-context demonstrations for the neural module functionality. $50$ samples are annotated as a small retrieve pool (shared across datasets) for mapping a column to a new one according to the question, $i.e.$, $f_m$ API in BINDER grammar. A simple BLEU score retriever will retrieve $8$ similar items from the pool as in-context demonstrations for each coming sample. Below are several examples demonstrating the formats of neural module functionality prompt. The output (sub-)table contains a new mapped column named after the query, which will be further merged into the original table. Note it is important to formulate the output also as a (sub-)table, otherwise if Codex outputs a list of answers, it will be confused which row it is generating answer for and perform very poorly.

```
## 1
Give a database as shown below:
Table: Highest mountain peaks of California
/*
row_id   prominence
0        10080 ft; 3072 m
1        1677 ft; 511 m
2        7196 ft; 2193 m
3        2894 ft; 882 m
4        9832 ft; 2997 m
5        2563 ft; 781 m
*/
Q: Answer question "What is the value of in feet?" row by row.
QA map@ output:
```

```
/*
row_id  prominence          What is the value of in feet?
0       10080 ft; 3072 m    10080
1       1677 ft; 511 m      1677
2       7196 ft; 2193 m     7196
3       2894 ft; 882 m      2894
4       9832 ft; 2997 m     9832
5       2563 ft; 781 m      2563
*/

## 2
Give a database as shown below:
Table: 2010-11 UAB Blazers men's basketball team
/*
row_id  hometown
0       chicago, il, u.s.
1       oklahoma city, ok, u.s.
2       montgomery, al, u.s.
3       greenville, ms, u.s.
4       birmingham, al, u.s.
*/
Q: Answer question "Is it from alabama?" row by row.
QA map@ output:
/*
row_id  hometown                Is it from alabama?
0       chicago, il, u.s.       no
1       oklahoma city, ok, u.s. no
2       montgomery, al, u.s.    yes
3       greenville, ms, u.s.    no
4.      birmingham, al, u.s.    yes
*/

## 3
Give a database as shown below:
Table: 1963 International Gold Cup
/*
row_id  driver
0       jim clark
1       richie ginther
2       graham hill
3       jack brabham
4       tony maggs
*/
Q: Answer question "What is his/her country?" row by row.
QA map@ output:
/*
row_id  driver          What is his/her country?
0       jim clark       scotland
1       richie ginther  united states
2       graham hill     england
3       jack brabham    australia
4       tony maggs      south africa
*/
```

## A.3  MAJORITY VOTE STRATEGY

Majority vote is widely used to ensemble multiple candidate answers. A simple implementation of majority vote is to select the answer that appears most often in candidates as the final answer. For example, given a candidate answer pool $C$ with $n$ answers, the output answer $a_{out}$ is derived by:

$$a_{out} = \arg\max_a(count(a)), a \in set(C) \tag{1}$$

where $count(\cdot)$ function counts the number of occurrences of the input answer in $C$, and $set(\cdot)$ function de-duplicates the input list. In our case, $n$ programs are generated per sample, and their execution results compose the candidate answer pool.

In experiments, we adopt two variants of majority vote strategies. The first one we call *answer-biased*. In TABFACT, we conjecture that it is difficult to validate a statement of multiple sub-statements as *entailed* (*i.e.*, answer 1) with SQL program, since a minor error in program will cause the execution to output *refuted* (*i.e.*, answer 0). In other words, we value it when an answer 1 occurs because it indicates the statement has a high probability of being *entailed*. Thus, we re-weight answers by assigning four votes for an answer 1, and one vote for an answer 0. The second is *program-biased*. We assign larger weights to BINDER language when it occurs in the generated programs because BINDER consists of small portion in few-shot prompt, and thus more generations will be in standard programming language. Specifically, we re-weight BINDER program with ten votes, and the others with one vote in WIKITQ and MMQA.

## A.4  BINDER GRAMMAR ADAPTION TO SQL

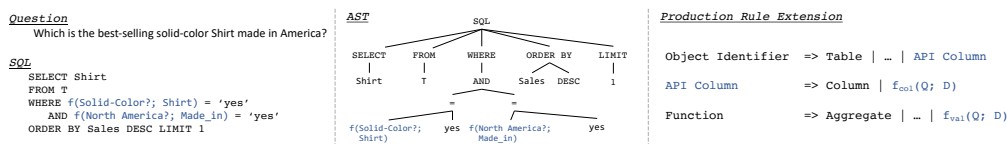

Figure 4: An illustration of BINDER program AST and extension to SQL grammar. The blue types in the production rule are extended by BINDER.

BINDER extends the production rules of the programming language to ensure the return values of API calls can be compatible with its original grammar. Take SQL as an example, in Figure 4, two APIs $f_{col}$ and $f_{val}$ are added as identifiers in the production rules to fit in as a *column* and a *value* in SQL. The AST of the shown BINDER program is quite similar to standard SQL's AST, except that API calls are placed in the column position.

## A.5  EVALUATOR

Program executions are naturally more difficult to match gold answers exactly when the answer is not a span in the input knowledge source, while end-to-end models may match this pattern by finetuning or infer phrases from the question. However, we find that in WIKITQ, some error cases of program execution outputs are correct according to humans. These cases mainly fall into two types. The first is *A or B* choice question, *e.g.,*, "Are there at least 13 different components on the chart?" in Table 9. The SQL program `SELECT COUNT(component) > 13`, returns 1 to indicate "yes" and 0 for "no". Generally in *A or B* problem, we match 1 with *A* and 0 with *B*. The second is *number with unit* question, *e.g.,*, "what is the difference in years between constituency 1 and 2" in Table 9, where the gold answer contains unit *years* in it, while it is almost impossible for programs to derive *years*, and 4 is also a reasonable correct answer in this case. Therefore, we add pre-matching logics upon WIKITQ official evaluator for these two problems. Besides, we also normalize the dates with $Recognizers$ package in Python. Table 9 gives some examples to show the differences of three WIKITQ evaluators. For fair comparison, all previous baselines are re-evaluated with the *semantic-match* evaluator in experiments.

| Example | EM(TaPEx string match) | EM(WIKITQ official) | EM(Semantic match) |
|---|:---:|:---:|:---:|
| (Normalization) *Question:* What was the same problem that Bernard Collomb had as Innes Ireland? *Gold Answer:* oil pressure *Pred Answer:* oil pressure (56 laps) | ✕ | ✓ | ✓ |
| (Float Precision) *Question:* What is the difference between the qualifying time in 1967 and 1965? *Gold Answer:* 7.45 *Pred Answer:* 7.449999999999989 | ✕ | ✓ | ✓ |
| (A or B Choice) *Question:* Are there at least 13 different components on the chart? *Gold Answer:* Yes *Pred Answer:* 1 | ✕ | ✕ | ✓ |
| (Number with Units) *Question:* What is the difference in years between constiuency 1 and 2? *Gold Answer:* 4 years *Pred Answer:* 4 | ✕ | ✕ | ✓ |

Table 9: Examples to illustrate differences among three exact match evaluators on WIKITQ. In experiments, we evaluate all baselines and our method with semantic-match evaluator.

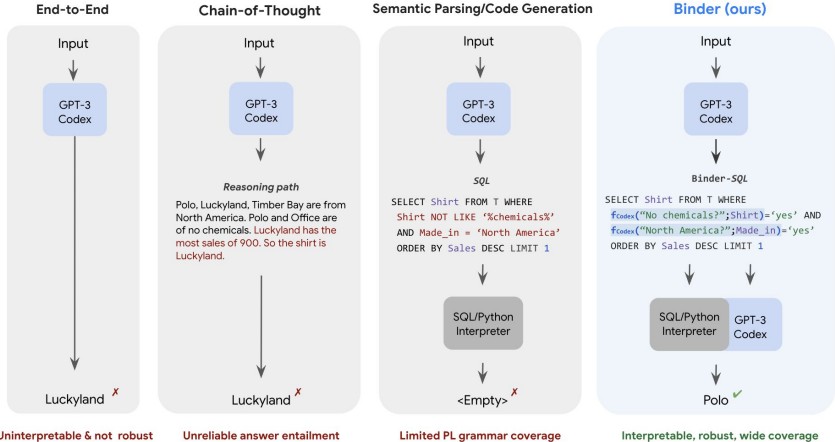

Figure 5: Comparison of the BINDER method(ours) with other large language model usage paradigms: End-to-End, Chain-of-Thought, and Semantic Parsing/Code Generation.

## B    COMPARISON WITH OTHER LARGE LANGUAGE MODEL USAGE PARADIGMS

Besides our work, some other works also focus on how to leverage large language models in creative ways to achieve better performance on downstream tasks and see improvements with proper designs. We compare our Binder method with the three previous paradigms of large language model usage: End-End, Chain-of-Thought, and Semantic Parsing/Code Generation.

**End-to-End**    End-to-End method (Brown et al., 2020; Hoffmann et al., 2022, *i.a.*) aims to use large language models to generate final answers directly, often done by providing a task description and/or a few examples for in-context learning. Despite being effective enough to reach state-of-the-art or comparable performance in a large number of benchmarks, it suffers from being uninterpretable and lacking in robustness.

**Chain-of-thought**    Chain-of-thought methods  (Wei et al., 2022; Chowdhery et al., 2022a; Kojima et al., 2022; Chung et al., 2022, *i.a.*)  improve the ability of large language models to perform complex reasoning by generating a series of intermediate reasoning steps. While achieving great success in various benchmarks, as a model-generated natural language, chain-of-thought suffers from unreliability and uncontrollability.

**Semantic Parsing/Code Generation**   Semantic Parsing/Code Generation methods (Chen et al., 2021; Rajkumar et al., 2022; Shi et al., 2022) aim to parse the question into a pre-defined program(SQL, Python, etc.), then execute it through the corresponding interpreter. It gains advantages in interpretability and robustness compared with End-to-End and Chain-of-Thought methods, but still suffers from the fixed grammar of the pre-defined programming language, making it inherently limited in coverage (Shi et al., 2020b).

**BINDER**   BINDER is a neural-symbolic paradigm that aims at mapping the question to a program that allows binding a unified LM API for additional functionalities. This keeps the interpretability and robustness of the semantic parsing method while unlocking the grammar's limitations on its coverage.

## C   MORE EXPERIMENTAL RESULTS

### C.1   RESULTS OF MORE PREVIOUS SYSTEMS

Due to page limits, we didn't list the complete previous systems in main results of WIKITQ and TABFACT. In this section, we present comparisons between more previous methods and ours, as shown in Table 10 and Table 11.

| Method | Dev | Test |
|---|---|---|
| *Finetuned* | | |
| Pasupat & Liang (2015) | 37.0 | 37.1 |
| Neelakantan et al. (2015) | 34.1 | 34.2 |
| Zhang et al. (2017) | 40.6 | 43.7 |
| Liang et al. (2018) | 42.7 | 43.8 |
| Dasigi et al. (2019) | 43.1 | 44.3 |
| Agarwal et al. (2019) | 43.2 | 44.1 |
| Wang et al. (2019) | 43.7 | 44.5 |
| Herzig et al. (2020) | - | 48.8 |
| Xie et al. (2022)[¶] | 51.9 | 50.6 |
| Yin et al. (2020) | 53.0 | 52.3 |
| Yu et al. (2020) | 51.9 | 52.7 |
| Guo et al. (2021) | 53.6 | 52.3 |
| Liu et al. (2021)[¶] | 60.4 | 59.1 |
| Zhou et al. (2022)[¶] | 61.1 | 61.3 |
| Jiang et al. (2022)[¶] | - | 63.3 |
| *Without Finetuning* | | |
| Codex end-to-end QA[¶] | 50.5 | 48.7 |
| Codex SQL[¶†] | 60.2 | 61.1 |
| **Codex BINDER [†] (Ours)** | **65.3** | **65.2** |

Table 10: WIKITQ execution accuracy of more previous systems. ¶ means the result has been re-evaluated by the semantic-match evaluator in Table 9. † means symbolic method that outputs intermediate languages.

| Method | Small Test |
|---|---|
| *Finetuned* | |
| Chen et al. (2019a)[†] | 68.1 |
| Zhong et al. (2020)[†] | 74.3 |
| Shi et al. (2020a)[†] | 74.2 |
| Yang et al. (2020)[†] | 76.2 |
| Ou & Liu (2022)[†] | 77.0 |
| Liu et al. (2021) (BART-Large) | 82.5 |
| Eisenschlos et al. (2020) | 83.9 |
| Xie et al. (2022) | 85.4 |
| Liu et al. (2021) (Tapex) | **85.9** |
| *Without Finetuning* | |
| Codex end-to-end QA | 72.6 |
| Codex SQL[†] | 80.7 |
| **Codex BINDER [†](Ours)** | **85.1** |
| with few-shot retriever | **86.0** |

Table 11: TABFACT accuracy on small test set of more previous systems. † means symbolic method that outputs intermediate languages.

### C.2   RESULTS WITH WIKITQ OFFICIAL EVALUATORS

We list BINDER performance evaluated under WIKITQ official evaluator in Table 12. As shown, there exist performance drops compared with semantic-match evaluator. The symbolic methods with SQL and BINDER see larger drops (about $2.5\%$) than end-to-end methods (about $1\%$) because the *A or B choice* questions and *number with units* answers (illustrated in Table 9) are judged as incorrect.

### C.3   ABLATION STUDY OF #MAJORITY VOTE CANDIDATES

In this section, we give an empirical study of how number of generations (*i.e.*, candidate BINDER programs) affects majority voting performance. As shown in Figure 6 and Figure 7, increasing the

| Method | Dev | Test |
|---|---|---|
| *Finetuned* | | |
| T5-3B (Xie et al., 2022) | 51.6 | 50.3 |
| Tapex (Liu et al., 2021) | 60.1 | 58.5 |
| TaCube (Zhou et al., 2022) | 60.9 | 60.8 |
| OmniTab (Jiang et al., 2022) | - | **62.9** |
| *Without Finetuning* | | |
| Codex end-to-end QA | 49.3 | 47.6 |
| Codex SQL[†] | 57.6 | 55.1 |
| **Codex BINDER [†] (Ours)** | **62.6** | **61.9** |

Table 12: WIKITQ execution accuracy using its official evaluator, with a normalizer to recognize date.

#candidates of majority vote effectively improves the performance on WIKITQ and TABFACT. Empirically, TABFACT requires more candidates to determine *entailed* or *refuted*. We haven't conducted experiments on WIKITQ with more than 20 candidate programs now, which may potentially lift the current best performance a bit.

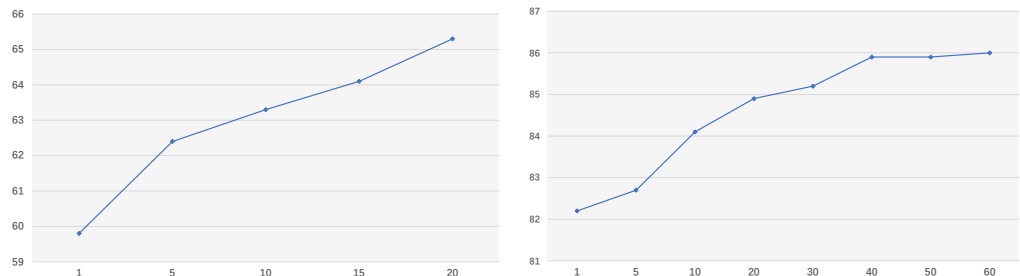

Figure 6: WIKITQ execution accuracy on dev set with different numbers of generations (x-axis) used in majority vote.

Figure 7: TABFACT execution accuracy on small test set with different numbers of generations (x-axis) used in majority vote.

### C.4 WIKITQ SQL UNSOLVABLE PROPORTION BY OUR STATISTICS

In Table 3, the decomposition of execution accuracy on WIKITQ development set, we can see the BINDER method also gain advantage on Program-solvable subset (and pure SQL method still able to have a certain performance on Program-unsolvable part). It is surprisingly since this part could already be solved by SQL (according to notation from SQUALL dataset), and thus our approach does not theoretically improve the performance of this part. Further, we found this is because in SQUALL, authors normalized part of the table before doing the SQL annotation, and they also missed some samples that could be solved after normalizing the table. In turn, there is a significant portion of the data marked by SQUALL, and when we use the original table rather than the table modified by the authors, we find that it belongs to the part that cannot be resolved by the program, so that BINDER could gain some advantage on it. The same is true for the part that is not marked by SQUALL. We re-executed the SQL annotation of the SQUALL dataset on the tables we normalized for each sample, check and re-judged whether the sample was a Program-solvable sample by whether the execution result matched the previous one, and obtained the new split of 1251 (44.19%) and 1580 (55.81%) for Program-unsolvable and Program-solvable set, respectively.(Previous split according to SQUALL is of 625 (22.08%) and 2206 (77.92%) examples. This reduces table content noise by artificially cleaning the table, but also underestimates the challenges posed by in-the-wild data.) The new split will be released in our code for future more fine-grained analysis or exploration.

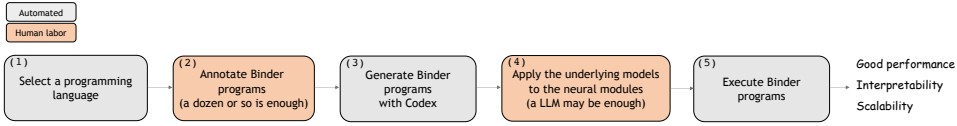

Figure 8: A pipeline to extend BINDER to a new domain.

## D  PIPELINE TO EXTEND BINDER

In this section, we introduce a pipeline with Codex to extend BINDER to a new domain with new the programming language and neural modules. Figure 8 illustrates the pipeline:

(1) A programming language is selected as the intermediate representation according to the domain. For example, SQL fits table domain, SPARQL, Cypher fit knowledge graph domain, and general-purpose languages like Python, Lisp, and Java are more likely to be adapted to various domains.

(2) Before parsing the text into BINDER programs, in-context examples are required to help Codex learn this task and the grammar adaption (inserted neural modules) from the original language. Empirically, since Codex has been heavily pretrained on code, a dozen of BINDER program annotations are enough. Besides, annotating a pool of more examples (*e.g.*, 100) and retrieving in-context examples for each inference sample has been proved to be a effective way to improve parsing accuracy, as we applied in TABFACT.

(3) Generate the BINDER programs with Codex given the in-context examples and the inference query. Codex hyper-parameters can be adjusted based on the task.

(4) Bind models with the neural module functionalities in BINDER program. In this paper, we set Codex as the only underlying model for all functionalities like commonsense QA, information extraction because large language models (LLM) are powerful and easy-to-use. If a single LLM is not enough to realize the required functionalities, *e.g.,* VQA, bind the API with a specialized VQA model given the inputs are images.

(5) Execute the BINDER programs with the deterministic language executor and neural models.

## E  ERROR ANALYSIS

### E.1  ERROR TYPES

We divide the error type into:

(1) **Syntax error**: prediction not satisfied with the syntax of BINDER program.

(2) **Semantic error**:
  - *Column*: choose the wrong column, or miss the restriction from certain column.
  - *Value*: use the wrong value, or obfuscate the fuzzy match with the exact match.
  - *Operator*: use wrong operator or miss the restriction of operators.
  - *Structure*: The query produced an overall structural error. Either the overall meaning of the query is misinterpreted; or the query only considers the question and is not applicable to this table at all.
  - BINDER *usage*: missing the usage of BINDER when needed, or the command inside the BINDER cell is improper or ambiguous to execute.

(3) **Incorrect execution**: The predicted program is correct, but the final result are wrong because some wrong mid-term answers are predicted in the execution process(error in information extraction, error in handling the corner cases value etc.).

(4) **False negative**: the annotation of the gold answer is wrong; or the prediction is actually right while the executor misjudges it as wrong due to limited ability.

| row_id | year | site | winning team | losing team | series |
|--------|------|------|--------------|-------------|--------|
| 0 | 1902 | lawrence | kansas 16 | kansas state 0 | ku 1–0 |
| 1 | 1903 | lawrence | kansas 34 | kansas state 0 | ku 2–0 |
| 2 | 1904 | manhattan | kansas 41 | kansas state 4 | ku 3–0 |
| 3 | 1905 | lawrence | kansas 28 | kansas state 0 | ku 4–0 |
| 4 | 1906 | manhattan | kansas state 6 | kansas 4 | ku 4–1 |
| 5 | 1907 | lawrence | kansas 29 | kansas state 10 | ku 5–1 |
| ... | | | | | |
| 21 | 1924 | manhattan | kansas state 6 | kansas 0 | ku 17–2–3 |
| 22 | 1925 | lawrence | kansas state 14 | kansas 7 | ku 17–3–3 |
| 23 | 1926 | manhattan | kansas state 27 | kansas 0 | ku 17–4–3 |
| 24 | 1927 | lawrence | kansas state 13 | kansas 2 | ku 17–5–3 |
| 25 | 1928 | manhattan | kansas 7 | kansas state 0 | ku 18–5–3 |
| ... | | | | | |

Table 13: The table of WIKITQ nt-1239 with title "Kansas–Kansas State football rivalry". We use it for demonstration of interpretibility. Removed rows unimportant to get the answer. The query is *When was the first game that kansas state won by double digits?* and the gold answer is *1926*.

## E.2 ERROR EXAMPLE

The context(table, query, answer) of WIKITQ nt-1239 is shown in Table 13.

## F ANNOTATION INTERFACE

We build an annotation interface allowing real-time executions with huggingface spaces[4]. Currently, SQL and BINDER with SQL are supported. We will release the source code which hopefully may benefit annotations of tasks requiring symbolic languages in the community, *e.g.*, semantic parsing and code generation. Figure 9 and 10 demonstrate our interface.

---

[4]https://huggingface.co/spaces

## Question: when was the last time kansas state lost with 0 points in manhattan?

## Title: Kansas–Kansas State football rivalry

|   | row_id | year | site | winning team | losing team | series |
|---|---|---|---|---|---|---|
| 0 | 0 | 1902 | lawrence | kansas 16 | kansas state 0 | ku 1–0 |
| 1 | 1 | 1903 | lawrence | kansas 34 | kansas state 0 | ku 2–0 |
| 2 | 2 | 1904 | manhattan | kansas 41 | kansas state 4 | ku 3–0 |
| 3 | 3 | 1905 | lawrence | kansas 28 | kansas state 0 | ku 4–0 |
| 4 | 4 | 1906 | manhattan | kansas state 6 | kansas 4 | ku 4–1 |
| 5 | 5 | 1907 | lawrence | kansas 29 | kansas state 10 | ku 5–1 |
| 6 | 6 | 1908 | lawrence | kansas 12 | kansas state 6 | ku 6–1 |
| 7 | 7 | 1909 | manhattan | kansas 5 | kansas state 3 | ku 7–1 |
| 8 | 8 | 1911 | manhattan | kansas 6 | kansas state 0 | ku 8–1 |
| 9 | 9 | 1912 | lawrence | kansas 19 | kansas state 6 | ku 9–1 |

## Answer: ['1964']

## Full info for this dataset item(if you need)

▸ {...}

Neural SQL

SELECT MAX(Year) FROM w WHERE QA("map@Did Kansas State lost with 0 points?"; `losing team`)=";

## Execution Answer: [1964]

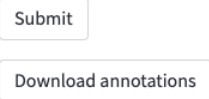

Figure 9: An example from WIKITQ annotation interface.

## Statement: the album powerhouse be release in 1968

## Table:

`caption`

`none`

|   | row_id | year | title | from album | label |
|---|--------|------|-------|------------|-------|
| 0 | 0 | 1962 | 2022-8-29tni / sinnin' sam | lookin' ahead | pacific jazz |
| 1 | 1 | 1962 | the young rabbits / song … | lookin' ahead | pacific jazz |
| 2 | 2 | 1962 | congolese sermon / weat… | (only on 45) | pacific jazz |
| 3 | 3 | 1963 | no name samba / tough talk | tough talk | world pacific |
| 4 | 4 | 1963 | turkish black / boopie | tough talk | world pacific |
| 5 | 5 | 1963 | spanish castles / bluesette | jazz waltz (w les mccann) | world pacific |
| 6 | 6 | 1964 | heat wave / on broadway | heat wave | world pacific |
| 7 | 7 | 1964 | i remember 2022-8-30 / lo… | stretchin' out | world pacific |
| 8 | 8 | 1965 | tough talk / the thing | the thing | world pacific |
| 9 | 9 | 1965 | aqua dulce / soul bourge | chile con soul | world pacific |

## Answer: 1

`1 means entailed, 0 means refuted.`

## Full info for this dataset item(if you need)

▸ `{...}`

Neural SQL

SELECT (SELECT year FROM w WHERE `from album` = "lighthouse '68") = 1968

## Execution Answer: [1]

Submit

Download annotations

Figure 10: An example from TABFACT annotation interface.

