# OpenReview forum: "Binding Language Models in Symbolic Languages"
_ICLR.cc/2023/Conference — ICLR 2023 notable top 25%_

### Official Review · Reviewer_S9TA · 2022-10-24

**Confidence:** 4
**Correctness:** 3
**Technical Novelty And Significance:** 3
**Empirical Novelty And Significance:** 3
**Recommendation:** 8

**Clarity, Quality, Novelty And Reproducibility:**

The BINDER system can benefit lots of semantic parsing tasks since it uses uniformed underlying API, i.e., SQL and Python, and requires on further training process.

**Strength And Weaknesses:**

Strength:
- The system does not require extra training but can achieve better performance compared to fine-tuning settings.
- The in-depth analysis validates that BINDER can have significantly better performance on program-unsolvable questions.

Weakness:
No obvious weakness was observed.

**Summary Of The Paper:**

This paper propose BINDER, a training-free neural-symbolic framework that (1) maps the task input to a program using a pre-trained language model, where the model has to decide which part in the input can be converted to a target programming language and corresponding tasks API calls for further extension; (2) adopts an LM as both the program parser and the underlying model called by the API during the execution, where a language model serves to realize the API calls given the prompt and the return values feed back into the original programming language; (3) derive the final answer via a deterministic program interpreter executing the program without API calls. BINDER achieves state-of-the-art results on WikiTableQuestions and TabFact datasets, where previous work is all fine-tuned on a large-scale training set.

**Summary Of The Review:**

- This paper is clear and well-organized. Lots of complementary materials are also provided in the Appendix to help us understand the technical details better.
- The methods can be applied to a broad range of semantic parsing tasks since it use uniformed underlying APIs, i.e., SQL and Python, and require no further training process.
- The experimental results are impressive. It outperforms previous work which typically use finetuning settings based on large training set.

---

> ### Author Response · Authors · 2022-11-13
> **Thank you for your review**
>
> Thank you for your appreciation and detailed evaluation of our work.
>
> We are glad the experimental details and other materials in Appendix can help the audience understand the technical details of our framework. We hope Binder can be a step towards making symbolic reasoning (e.g., semantic parsing) more robust and capable. We will help extend that to more semantic parsing tasks and verify this framework in other tasks.
>
> We are happy to see you describe the result of our attempt as “impressive”. The large improvement of Binder to leverage Codex in a neural-symbolic way is indeed surprising. We will further explore the upper bound of Large Language Models in code generation tasks and beyond.

---

### Official Review · Reviewer_8V6D · 2022-10-25

**Confidence:** 3
**Correctness:** 4
**Technical Novelty And Significance:** 4
**Empirical Novelty And Significance:** 4
**Recommendation:** 8

**Clarity, Quality, Novelty And Reproducibility:**

Clarity is good.

Quality is good.

Novelty is good.

Reproducibility is good as code is attached.

**Strength And Weaknesses:**

This paper is a creative usage of a code generation model. It also represents an interesting new approach to neural symbolic: code generation models can learn to divide a task into a high-level symbolic logical program and some low-level components for neural inference.

The experimental results are promising. With a pre-trained Codex model, without fine-tuning and with a small number of annotated examples, the proposed method is able to achieve state-of-the-art on two QA datasets. It achieves the desired behavior of a neural symbolic method, including better interpretability and answering some difficult questions.

**Summary Of The Paper:**

This paper proposes a training-free two-stage method based on Codex model for QA tasks. The first stage uses few-shot learning to prompt Codex and converts a natural language question to a SQL or Python program, where difficult-to-resolve parts are represented as API calls. In the second stage, Codex is prompted again to generate answers to each said API call, and then the whole program is executed to obtain the answer. Experimental results show new state-of-the-art on WikiTQ and TabFact datasets, with particular performance gains on questions that are unsolvable by natural-language-to-SQL translation.

**Summary Of The Review:**

The main idea is good and well executed.

---

> ### Author Response · Authors · 2022-11-13
> **Thank you for your review**
>
> Thanks for your appreciation of our work and careful review!
>
> We are honored to see you describe our attempt as “creative” and “interesting”. It is exciting to see LLMs like GPT3 Codex can predict such a never-existing program we created and can learn to divide a task into a high-level symbolic logical program and some low-level components for neural inference. It also indicates that proper usage could raise the upper bound of a language model.
>
> We also believe a proper combination of neural approach and symbolic approach is a promising direction for good performance, higher interpretability, and robustness. Our analysis of Binder showed it achieved the desired behavior on the aspects above using only dozens of annotations. In the future, we will continue to explore Binder under more tasks and new scenarios.

---

### Official Review · Reviewer_Pz8D · 2022-10-27

**Confidence:** 2
**Correctness:** 3
**Technical Novelty And Significance:** 3
**Empirical Novelty And Significance:** 3
**Recommendation:** 8

**Clarity, Quality, Novelty And Reproducibility:**

Clarity needs to be improved. Quality, Novelty And Reproducibility seem to be good.

**Strength And Weaknesses:**

Strength:
1. Incorporating symbolic component into end2end learning systems is always an interesting idea
2. The proposed approach achieved state-of-the-art results

Weakness:
1. The writing is not clear throughout.  Most importantly, the paper directly jumps in talking about the proposed method without first clearly introducing the problem setup.
2. Again, presentation is not clear.  The main figure is confusing.  I do not understand how the prompting part is employed exactly in the system.  In particular, it is not clear how the prompting results are combined with the parsed results to feed in the program interpreter.
3. The results are a bit over sold.  Even though the fine-tuning methods require more task annotations, but GPT3 codex is pretrained on many more data.  The improvement is not that surprising.

**Summary Of The Paper:**

1) The paper proposed BINDER to incorporate symbolic component into large language models, and the main benefit of the method is not requiring any fine-tuning.
2) BINDER outperforms state-of-the-art results on WIKITABLEQUESTIONS and TABFACT datasets
3) useful analysis is presented to help readers understand the proposed approach.

**Summary Of The Review:**

The proposed idea is exciting, the results are fairly strong, but the writing needs to be improved.
[Edited: The author has improved their writing to make the paper much clearer.  I have thus raised my score from 6 -> 8.]

---

> ### Author Response · Authors · 2022-11-13
> **Thank you for your review**
>
> Thank you for your kind feedback and valuable comments. We have revised our manuscript and addressed several points you mentioned:
>
> **W1: The writing is not clear throughout. Most importantly, the paper directly jumps in talking about the proposed method without first clearly introducing the problem setup.**
>
> We have added a separate paragraph *Task Definition* (purple color in the pdf) to describe the problem setup Binder targets at the beginning of Section 2 (Approach) in the new manuscript. In the previous version of the paper, problem setup was introduced along with the Binder framework in a single *Overview* paragraph (Section 2.1), which might be unclear to notice. And now we have separated them apart to be more explicit. Specifically, the target tasks accept a question/statement Q and optional contexts(e.g., passage, table, kb) as inputs, and output an answer A that responds to Q. Instead of predicting A in an end-to-end manner, Binder first converts Q to an executable Binder program Z and then executes Z to derive the answer A.
>
> **W2: Again, presentation is not clear. The main figure is confusing. I do not understand how the prompting part is employed exactly in the system. In particular, it is not clear how the prompting results are combined with the parsed results to feed in the program interpreter.**
>
> We have updated Figure 1 in the new manuscript. We have revised Figure1 to present a more detailed Binder execution process. Specifically,  for *f(“North America?”; Made_in)* API call, a question(prompt) is asked on each value in *Made_in* column using Codex, and the output is a new column *North_America* with yes/no values (the column naming follows the question). Next, the *North_America* column replaces the API call *f(“North America?”; Made_in)* in SQL, which is now a standard SQL and can be fed into a program interpreter.
>
> **W3: The results are a bit over sold. Even though the fine-tuning methods require more task annotations, but GPT3 codex is pretrained on many more data. The improvement is not that surprising.**
>
> Firstly, since Binder-program is a new kind of program that never existed before, it is not likely for LLMs like GPT-3 Codex to see them during pre-training. Though Codex is familiar with SQL/Python, we find it interesting that it learns Binder grammar adaption with a few in-context exemplars, i.e., dividing a task into a high-level symbolic logical program and some low-level components for neural inference.
>
> Secondly, we do agree LLMs are pretrained to be powerful (maintain the lower bound), but the performance of LLMs also largely depends on how we use them (explore the upper bound). In Table 1 & 2, we already show that simply using Codex in an end-to-end QA manner falls far behind Codex Binder (-15.9% and -13.4% on WikiTQ and TabFact) and previous SOTA finetuned systems (-14.6% and -13.3%). We add a new section in Appendix B to summarize and compare methods that leverage LLM in creative ways and achieve promising performance on down-stream tasks. We think the large improvement of Binder to leverage Codex in a neural-symbolic way is non-trivial~(as R2 commented as creative) in this sense.

---

> > ### Comment · Reviewer_Pz8D · 2022-11-15
> > **Thanks for the revision**
> >
> > Thanks for addressing my comments.  The paper becomes much clearer to me.  The only remaining point I have about the task setup is when you said "accept a question/statement Q and optional contexts(e.g., passage, table, kb)", is it more precise to frame it as a question / statement Q with a structured context?  I don't see how BINDER applies to passages, and the experiments were not done on unstructured contexts either.

---

> > > ### Author Response · Authors · 2022-11-16
> > > **Thank you for your kind and timely reply**
> > >
> > > Thank you for your kind and timely reply.
> > >
> > > The Binder framework supports contexts like unstructured text. In Figure1, the *“No chemicals?”* sub-question in API call actually queries the textual passages, i.e., the *Details* tags, linked to the *Shirt* column in the table since the compositions of each shirt are listed in the textual passages rather than the table cells.
> > >
> > > In experiments, we tested Binder on the MultiModalQA dataset in Section 4.4 (Binder Extension), a multi-modal QA task with contexts across tables, text, images, and combined (for images, we preprocess them into texts with an image-caption model). Binder can achieve better performance than baselines and is comparable with sota finetuned methods.
> > >
> > > Specifically, if the context is composed of passages linked with table cells, the passages are filled into the table cells and then handled by a Binder-SQL program. If the context is the pure passage(s), the Binder program is a *f$_{val}$(question; passage_title)* API call as described in Section 2.3 (Binder Implementation). In future work, other symbolic languages (maybe domain-specific languages) more suitable to passages and images can be explored so that the reasoning steps over them can be more fine-grained.

---

### Author Response · Authors · 2022-11-13
**Summary response to all reviewers and the new revision**

We sincerely thank all the reviewers for their feedback and constructive comments. We are pleased that the reviewers appreciate our work and find it exciting/creative and well-executed/beneficial (R1, 2, 3), the experiments strong/promising/impressive (R1, 2, 3), the analysis useful/in-depth (R1, 3), and our paper well-organized (R3).

We’ve revised and updated the draft, which reflects the reviewers' comments (all revisions are highlighted in purple color in the new pdf). The updates are summarized as follows:
- Section 2: add a separate paragraph Task Definition to describe the problem setup Binder targets. (R1)
- Figure 1: revise to present a more detailed Binder execution process. (R1)
- Section 1, 2.1, 2.2: add more detailed explanations of the Binder execution process. (R1)
- Appendix B: add a section to summarize and compare the methods that leverage large language models in creative ways. (R1)

---

### Decision · Program_Chairs · 2023-01-20

**Decision:**

Accept: notable-top-25%

**Justification For Why Not Higher Score:**

The main setting of the paper, question answering with a database, is still relatively narrow.  It would be more impactful if the targeted domain is more general.

The clarity of the paper can be improved, at the moment the paper is not the easiest to understand.

**Justification For Why Not Lower Score:**

The paper is well executed, with interesting ideas and good experiment results and no major problems.  These factors distinguish this paper from other normal accepted papers.

**Metareview: Summary, Strengths And Weaknesses:**

This paper presents an interesting approach of solving question answering problems by generating a program first and then executing it to get answers.  The programs are generated by few-shot prompting a large language model (LLM).  The difficult to resolve part of the programs are delegated again to an LLM through the special LLM API calls, that are again resolved with few-shot prompting.  The inclusion of API calls used in the programs to query LLMs is a particularly interesting idea to combine programmatic solutions with neural models.  Results on a couple datasets show that this approach reaches state-of-the-art performance without additional training.

All reviewers liked the paper and recommended acceptance unanimously.  The clarity of the paper can still be improved, but overall this paper is well executed.

**Note From Pc:**

if the above contains the word "oral" or "spotlight" please see: "oral" presentation means -> notable-top-5% and "spotlight" means -> notable-top-25%. As stated in our emails, we are disassociating presentation type from AC recommendations